# Quantifying the Training Loads and Corresponding Changes in Physical Qualities among Adolescent, Schoolboy Rugby League Players

**DOI:** 10.3390/sports12090251

**Published:** 2024-09-12

**Authors:** Michael A. Carron, Aaron T. Scanlan, Thomas M. Doering

**Affiliations:** 1School of Health, Medical and Applied Sciences, Central Queensland University, Rockhampton 4701, Australia; a.scanlan@cqu.edu.au (A.T.S.); t.doering@cqu.edu.au (T.M.D.); 2St Brendan’s College Yeppoon, Rockhampton 4703, Australia

**Keywords:** pediatric, periodization, development, education, football, students

## Abstract

Objectives: The adolescent development period is critical for rugby league athletes, given the physical growth, neuromuscular adaptation, and skill acquisition that occurs. Secondary schools play an important role in the development of adolescent rugby league players; however, players may be selected into rugby league academies and development programs outside of school, as well as participating in additional sports. In turn, the training loads these young athletes accrue and the implications of these loads are currently unknown. Our aim was to quantify the training loads and concomitant changes in physical qualities of schoolboy and adolescent rugby league players during mesocycles within the pre-season and in-season phases. Design: This is a prospective experimental study. Methods: Twenty-one schoolboy rugby league players (16.2 ± 1.3 years) were monitored across separate 4-week mesocycles in the pre-season and in-season. Session frequency, duration, and the session rating of perceived exertion (sRPE) load were reported for all examples of training and match participation in the school rugby league program, as well as club and representative teams for any sport and personal strength and conditioning. Various physical qualities were assessed before and after each 4-week mesocycle. Results: The sRPE load that accumulated across the 4-week mesocycles was higher in the pre-season than the in-season (8260 ± 2021 arbitrary units [AU] vs. 6148 ± 980 AU, *p* < 0.001), with non-significant differences in accumulated session frequency and duration between phases. Session frequency, duration, and sRPE load differed (*p* < 0.05) between some weeks in an inconsistent manner during the pre-season and in-season mesocycles. Regarding physical qualities, improvements (*p* < 0.05) in the 10 m sprint test, Multistage Fitness Test, medicine ball throw, and 1-repetition maximum back squat and bench press performances were evident across the pre-season mesocycle, with declines (*p* < 0.05) in the 505-Agility Test, L-run Test, and 1-repetition maximum back squat performances across the in-season mesocycle. *Conclusions*: These novel training load data show schoolboy rugby league players experience considerable demands that may be suitable in developing several physical qualities during the pre-season but detrimental to maintaining such qualities across the in-season.

## 1. Introduction

Adolescent rugby league players form a critical part of the player development pathway to ensure the continuance and expansion of elite rugby league competitions. Accordingly, professional rugby league clubs and governing bodies, including the National Rugby League (NRL) [1], invest in junior academies and development programs targeting adolescent players to best prepare them for a successful, professional career in this sport [2]. In this regard, secondary schools can play a key role in aiding the development of adolescent rugby league players, given they deliver programs with focused rugby league training via specialized coaching, sport-specific strength and conditioning plans, and high-level schoolboy competition [3]. Schoolboy and adolescent rugby league players are at a particularly important time in their athletic development. Given the potential for adolescents to progress to professional ranks, Australian organizations like the Queensland Rugby League [2] support school systems that can have a considerable influence on the development of players. Indeed, with ~240,000 students engaging with the rugby league in the Australian school setting [1], school systems have the capacity to influence a considerable number of developing players.

An important consideration among schoolboy rugby league players is that they may participate in additional club and representative rugby league teams, as well as be selected into rugby league academies and development programs outside of the school setting [4]. In turn, schoolboy rugby league players may also participate in other sports as part of school, social, club, or representative teams [5]. In completing these holistic sporting commitments, schoolboy rugby league players likely participate in multiple training sessions and matches in conjunction with their school-based rugby league programs, potentially exposing them to excessive training loads. 

The training load is multifactorial and embodies acute and cumulative demands of training sessions and competition imposed on players over a set period [6]. As such, rugby league practitioners monitor the training loads experienced by their players to optimally prescribe and evaluate training plans alongside competitive requirements across different seasonal phases [7]. While there are several methods available to monitor the training load, many popular approaches, such as heart rate and global positioning system technologies, are not readily accessible to rugby league practitioners working in school settings due to practical constraints (e.g., financial cost, expertise, and labor, and difficulties tracking all activities among schoolboy players). Consequently, simple, feasible training load monitoring approaches, such as session frequency, duration, and the rating of the perceived exertion (sRPE) load holds strong utility for application in secondary school rugby league programs. 

To date, few studies have reported holistic training load data among adolescent athletes participating in school-based sports programs [4,8]. For instance, weekly sRPE loads were monitored in 75 adolescent, schoolboy rugby union players categorized into those only participating in the school program (15.2 ± 0.6 years, 2372 ± 1009 arbitrary units [AU]), and those completing additional representative (15.6 ± 0.7 years, 3645 ± 1588 AU) or talent squad commitments (16.4 ± 1.5 years, 2907 ± 1586 AU) (data calculated from published figures via WebPlotDigitizer, version 4.6) [8]. Based on these data, the authors concluded that performance may be compromised due to the high training loads encountered from extensive school and extracurricular sports participation among schoolboy rugby union players within an Australian secondary school context [8]. Moreover, Scantlebury et al. [4] documented weekly sRPE loads among representative (1378 ± 85 AU to 1784 ± 62 AU across terms) and non-representative (997 ± 46 AU to 1527 ± 66 AU across terms) (data calculated from published figures via WebPlotDigitizer, version 4.6) male and female players from various sports at a secondary school in the United Kingdom, but did not differentiate loads according to sport. As such, the training loads experienced by players participating in Australian rugby league school programs are yet to be quantified in the literature. Accordingly, schoolboy rugby league players may be exposed to excessive training loads, especially if undertaking additional rugby league and wider sports commitments, such as those reported in other school sports programs [4,8]. Indeed, these excessive training loads may promote undesired effects on the performance [9] and health of schoolboy rugby league players [10,11], which may lead to the discontinuation of the sport. Consequently, an analysis of the training loads experienced by adolescents and schoolboy rugby league players is needed to understand the demands imposed on these developing athletes and subsequent impacts to performance. 

Therefore, the aims of this study are to quantify the training loads and corresponding changes in performance during tests assessing physical qualities in adolescent schoolboy rugby league players during pre-season and in-season phases.

## 2. Methods

### 2.1. Study Design

A prospective experimental research design was followed. Training load was monitored across all rugby league players participating in a school rugby league program across two separate 4-week mesocycles in the pre-season and in-season phases of the annual season. The monitored pre-season mesocycle comprised the last 4 weeks (out of a 6-week pre-season phase), while the monitored in-season mesocycle comprised the last 4 weeks (out of a 12-week in-season phase) and contained the final four games of the competitive schoolboy rugby league season. Players’ self-reported completion of any training sessions or competitive matches during the monitoring periods included field training, strength and conditioning, and matches within the school rugby league program, training and matches for club or representative teams in rugby league or other sports, personal strength and conditioning, and physical education classes. To assess any corresponding changes in physical qualities surrounding each monitored mesocycle, a testing battery was administered the week before and the week after each mesocycle. 

### 2.2. Participants and Procedures

Male, adolescent rugby league players (*n* = 21; age: 16.2 ± 1.3 years; height: 180.2 ± 5.4 cm; body mass: 83.7 ± 10.3 kg) were recruited from the same male secondary school for this study. Players were eligible to participate if they competed in the first grade senior rugby league competition for the school, were free from injury and illness during monitored periods, and had completed at least six months of rugby league training immediately before study commencement. Written informed assent and consent were obtained from all players and their legal guardians prior to participation. Ethical approval for this study was obtained from the institutional Human Research Ethics Committee (#0000023570).

### 2.3. Training Load Monitoring

Each player documented all training sessions and matches they completed either in person to the lead researcher or electronically via personal mobile devices each day. For each session, players reported the activity as well as the duration across which they participated from the start of their structured warm-up to the completion of their structured cool-down, final repetition, or dismissal from coaching staff, depending on the type of session being completed. Scheduled durations set by the coaching staff were taken as the duration of training sessions administered in the school program following confirmation of their accuracy with the head coach. Alongside this information, players were instructed to report their personal RPE using the modified Borg category ratio-10 scale directly to the lead researcher for each session or match within 30 min of completion where possible but no longer than 24 h later [4], which supported validity in adolescent team sport players [12]. Players were familiarized with reporting RPE in a formal education session according to established recommendations [13], and RPE was also gathered from players in practice for 2 weeks before study commencement. Daily reminders were prompted to report RPE in person or via a message sent to their personal device. Training load data were tabulated for each week within each 4-week mesocycle and across the entire 4-week mesocycle in each phase.

### 2.4. Tests Used to Assess Physical Qualities

The most commonly adopted tests and testing protocols to assess physical qualities among adolescent rugby league players, as identified in a recent systematic review [14], were implemented on each testing occasion at the baseline and following the 4-week mesocycle in each phase. Tests assessing anthropometric qualities were initially administered on each occasion, whereby stature was measured using a stadiometer (SECA 213, SECA Corp; Hamburg, Germany) with players standing upright with their head positioned in the Frankfurt plane. Second, body mass was measured using electronic scales (SECA 813, SECA Corp; Hamburg, Germany) with players wearing normal rugby league training attire without shoes and socks. Finally, Σ4 skinfold thickness was taken across the biceps, triceps, subscapular, and supra-illiac landmarks [7] using Harpenden calipers (John Bull, British Indicators Ltd., England) following guidelines from The International Society for the Advancement of Kinanthropometry [15]. 

### 2.5. Linear Speed

Linear speed was assessed using the 10 and 20 m sprint times taken during 20 m linear sprints. Players were positioned with the toes of their leading foot placed 50 cm behind the first timing gate in a split stance before initiating each sprint [16]. Single-beam electronic timing gates (SmartSpeed Pro, Fusion Sport; Brisbane, Australia) were used to record sprint times to the nearest 0.001 s, with gates set at 0, 10, and 20 m. Players completed three attempts and were given 60 s of passive standing rest between attempts, with the best attempt for each distance recorded for analysis.

### 2.6. Change-of-Direction Speed

Change-of-direction (COD) speed was assessed using performance times taken during the 505-Agility Test and L-run Test. During the 505-Agility Test, players commenced each attempt with the toes of their leading foot on marking tape at 0 m. Players then moved as quickly as possible for 15 m in a forwards direction before turning 180° back toward the starting point for a further 5 m [17]. A single-beam electronic timing gate was positioned 10 m from starting point to record performance time to the nearest 0.001 s across the 5 m directly before and after the turn. 

During the L-run Test, players were positioned with the toes of their leading foot placed 50 cm behind the first timing gate in a split stance before initiating each attempt [16]. Players commenced each attempt on their own volition, moving as quickly as possible to a dome marker positioned 5 m in front, before turning 90° in the designated direction and moving to and around another dome marker positioned 5 m away and returning on that same course back to the starting point, while always moving in a forward direction. A single-beam electronic timing gate was positioned at 0 m to record the performance time to the nearest 0.001 s. The test consisted of a first turn at 90°, second turn at 180°, and final turn at 90°. For both tests, players completed three attempts in each direction with 60 s of passive standing rest between attempts, with the best attempt in each direction recorded for analysis.

### 2.7. Aerobic Capacity

Aerobic capacity was estimated using the MSFT following established procedures [18]. The MSFT involves repeated 20 m shuttle runs with progressive increases in speed dictated by audio cues. The MSFT was concluded when players either voluntarily stopped or failed to cover the requisite distance, as signaled by audio cues across two successive shuttles. Maximal oxygen uptake (VO_2max_ [mL·kg^−1^·min^−1^]) was estimated from the last successful shuttle completed during the MSFT using a valid equation [18].

### 2.8. Maximal Muscular Power

Maximal upper-body and lower-body muscular power were assessed via the medicine ball throw (MBT) and countermovement jump (CMJ), respectively. The MBT required players to throw a 2 kg medicine ball (Celsius; Melbourne, Australia) horizontally to achieve the maximum distance (m) possible. Players were seated on a flat bench with their backs in contact with an adjacent wall, knees flexed to 90°, and feet in full contact with the floor. A measuring tape was fastened to the floor at the wall positioned directly below the player when sitting on the bench with distance measured in a direct line from the wall to the initial floor marking (indicating the landing of the ball, which was covered in chalk) to the nearest 0.1 cm. 

The CMJ required players to jump in a vertical direction and displace the highest vane possible on a yardstick device (Swift; Lismore, Australia). Players self-selected their preferred squat depth during the countermovement before jumping and utilized an arm swing during jumping, with CMJ height measured to the nearest 1 cm. For both tests, players completed three attempts with 60 s of passive standing rest between attempts, with the best attempt recorded for analysis.

### 2.9. Muscular Strength

Muscular strength was assessed using the 1-repetition maximum (RM) test for the bench press, back squat, and prone row exercises in this order. For each exercise, procedures stipulated by the National Strength and Conditioning Association [19] were employed, with a maximum of five attempts permitted with 180 s of seated rest between attempts [19]. All attempts were completed using a 2.13 m Olympic bar (20 kg). The bench press was assessed with hands pronated using a hook grip and players maintaining a supine position on a flat bench. The back squat was performed with self-selected footing using a high position and a pronated hook grip, with players required to reach a depth where their thighs were parallel to the floor. The prone row was performed with a pronated hook grip, with players laying in a prone position on a flat bench and pulling the bar towards their body until contacting the underside of the bench (12 cm thickness). The highest 1-RM attempt (kg) for each exercise was recorded for analysis. 

### 2.10. Statistical Analysis

Normality and homogeneity of variance were confirmed for all variables using Shapiro–Wilk and Levene’s tests, respectively. All descriptive data for variables were reported as the mean ± standard deviation (SD). Paired *t*-tests were used to assess differences between total session frequencies, durations, and sRPE load across each 4-week mesocycle, while separate repeated-measure one-way analyses of variance with Tukey’s post hoc tests were used to assess differences in training load variables between weeks within each 4-week mesocycle. Paired *t*-tests were used to assess differences in the physical qualities measured before and after each mesocycle. Magnitudes of differences for pairwise comparisons were determined using Cohen’s d_av_ effect size calculated using a published spreadsheet [20] (with 95% confidence intervals [CI]) and interpreted as follows: *trivial*, <0.20; *small*, 0.20–0.49; *medium*, 0.50–0.79; or *large*, ≥0.80 [21]. All analyses were conducted using PRISM software (version 10.1.1 (270); GraphPad Software, Boston, MA, USA) and Microsoft Excel (version 15; Microsoft Corporation, Redmond, WA, USA).

## 3. Results

Descriptive data for, and comparative statistics between, the 4-week mesocycles in each phase, which are shown in Table 1. Non-significant, *small* differences in session frequency and duration were evident between phases. However, a significantly *larger* sRPE load was evident in the pre-season compared to the in-season. 

Figure 1 shows group and individual data for session frequency, duration, and sRPE load each week during each mesocycle. Significant differences between weeks within the pre-season were apparent for session frequency (*p* < 0.001) and duration (*p* = 0.002), but not sRPE load (*p* = 0.053). Post hoc analyses revealed that the session frequency and duration in week 2 was significantly higher (*p* ≤ 0.001–0.036, *large* effects) than all other weeks. Significant differences between weeks within the in-season were found for session frequency (*p* < 0.001), duration (*p* < 0.001), and sRPE load (*p* < 0.001). Post hoc analyses revealed that session frequency was significantly lower in week 1 (*p* < 0.001–0.019, *large* effects) than all other weeks; session duration was significantly lower in week 1 than weeks 2 and 3 (*p* ≤ 0.001–0.031, *large* effects), and significantly higher in week 2 than weeks 3 and 4 (*p* ≤ 0.001–0.012, *large* effects); and sRPE load was significantly lower in week 1 than weeks 2 and 3 (*p* < 0.001–0.003, *large* effects), as well as significantly lower in week 4 than weeks 2 and 3 (*p* < 0.001–0.010, *large* effects).

Table 2 provides descriptive data for, and comparative statistics between, physical qualities before and after each 4-week mesocycle. Significant (*p* < 0.001–0.002, *large* effects) improvements in the 10 m sprint test, MSFT, MBT, 1-RM back squat, and 1-RM bench press performances were evident across the pre-season mesocycle. In contrast, significant (*p* = 0.002–0.032, *small*–*medium* effects) declines in the 505-Agility Test, L-run Test, and 1-RM back squat performances were evident across the in-season mesocycle.

## 4. Discussion

This study is the first to quantify the training load experienced by adolescent, schoolboy rugby league players alongside corresponding changes in performance indicated by testing physical qualities. The main findings were as follows: (1) players experienced significantly higher sRPE loads, but similar session frequencies and durations were found during the pre-season compared to in-season mesocycle; (2) inconsistent differences in training loads were evident from week-to-week across variables in the pre-season and in-season mesocycles; and (3) significant pre-season improvements and in-season decreases were observed in several physical qualities. 

This study is the first to quantify training loads in schoolboy rugby league players, demonstrating that they completed 6.6 ± 2.0 sessions, for 334 ± 103 min, with 1800 ± 610 AU per week on average across both monitoring periods. These weekly data exceed those previously reported in non-representative players (5.2 ± 0.6 sessions and 1271 ± 205 AU per week) but are comparable to those reported in representative players (6.4 ± 1.1 sessions and 1597 ± 165 AU per week) among adolescent males and females competing in various school sports within the United Kingdom [4]. Moreover, the weekly sRPE load data we observed were less than those reported in adolescent male rugby union players (3645 ± 1588 AU per week) [8]. Consequently, the collective evidence suggests the training loads experienced for players competing in school programs may vary, potentially due to the sporting code involved as well as the varied additional requirements experienced, such as participating in representative teams. Interestingly, the weekly sRPE load data we observed are comparable to those documented in professional male rugby league players across a season (1687 ± 28 AU [22]), suggesting that they may be excessive for an adolescent schoolboy population. Nevertheless, we cannot definitively determine whether the weekly loads we observed are appropriate, given the lack of supported guidelines available for adolescent schoolboy rugby league. Indeed, the Queensland Rugby League (QRL), a state-based organization within Australia, recommend that 16-year-old rugby league players can participate in up to 11 sessions and 390 min per week in the pre-season, and 11 sessions and 420 min per week during the in-season within a club setting [2], which exceed the weekly session frequencies and durations we observed. However, there is no sound rationale nor evidence cited within the current QRL framework, making it unclear how they were derived. Furthermore, solely recommending weekly session frequencies and durations does not account for the intensities encountered, which are paramount to consider given their role in inducing positive adaptations [10,23] and contributing to injury risk [6,7] among rugby league players. Consequently, recommendations are needed from professional bodies to better guide the loading prescription within adolescent rugby league players. 

When considering fluctuations in weekly loads within each phase, we showed a negligible variation from week-to-week during the pre-season mesocycle with more variation in weekly loads during the in-season mesocycle. These weekly trends suggest that a limited periodization strategy was experienced by players within each phase given that we might expect weekly load to incrementally increase across the pre-season as the goal of this phase is generally to induce positive adaptations with progressive training stimuli [24], and then remain somewhat consistent during the in-season to reduce injury risk [25] and maintain the developed physical qualities [26]. However, these findings may be attributed to some players concurrently participating in various rugby league and other sports programs, teams, or representative duties, making it difficult for coaching staff, both within and outside of the school program, to effectively prescribe the intended periodization schemes in light of the holistic demands their players’ experience. Consequently, school coaches working with adolescent players must consider that they are likely to participate in training and compete for multiple teams across different settings, elevating their exposure to training and match stimuli compared to what is planned solely within the school program. Therefore, communication between school team coaching staff and external team coaching staff (e.g., academy or representative teams), alongside suitable monitoring procedures adopted in all environments, is necessary to understand the complete demand experienced and, in turn, prescribe training loads that are appropriately periodized among schoolboy rugby league players [4].

While weekly load data did not demonstrate expected trends within each phase, the accumulated training load across mesocycles revealed some notable trends between phases. Specifically, session frequency and duration were relatively consistent, signifying that players may have had similar exposures between phases, but the sRPE load was significantly higher in the pre-season, demonstrating a more intensified approach during this phase. In line with our findings, other studies in professional, male rugby league players have consistently shown that a higher sRPE load is encountered during the pre-season than the in-season [25,27]. In this regard, coaches may strategically plan elevated training loads during the pre-season to elicit desirable biological adaptations in key physical qualities [24]. In turn, lower loads may be prescribed in the in-season to optimize readiness for competition each week and reduce the risk of sustaining injuries [28] or overtraining [29]. While the observed trends in accumulated training loads across the pre-season and in-season mesocycles may be expected, it is unknown whether they are appropriate in adolescent, schoolboy rugby league players given the limited available guidelines, which only stipulate weekly session frequency and duration recommendations [2]. Accordingly, the assessment of concomitant changes in performance using standardized testing around loading cycles may be useful to understand whether accumulated training loads may be appropriate [9,27]. 

In examining changes in performance before and after each mesocycle, we observed improvements in testing outcomes for several physical qualities across the pre-season, suggesting that loading may have provided suitable stimuli to elicit positive performance adaptations. In contrast, we observed declines in testing outcomes for several physical qualities across the in-season, suggesting that the loading extending into this phase may have been excessive and potentially maladaptive in nature. Indeed, the simultaneous requirement of training sessions and matches within and outside of the school-based rugby league program may be acutely sustainable, but when compounded chronically over multiple weeks and months, it likely explains the trends in performance that we observed. For example, research examining professional, male rugby league players shows that intensified training produces significant declines in aerobic capacity during a 7-week specific preparatory phase following 10 weeks of general preparatory training [9]. Moreover, higher training loads later during the in-season phase have been suggested to compromise physical development and recovery [30] while increasing injury incidences and fatigue in adult rugby league players [28]. Consequently, our study is the first to present training load and testing data (of physical qualities) in combination for adolescent, schoolboy rugby league players, suggesting that the development of key physical qualities may be achieved in the pre-season, but loads during the in-season may not permit further the improvement nor maintenance of testing performances within this sample and setting.

While this study is the first to quantify the training loads of schoolboys and rugby league players, it is not without its limitations. Firstly, the sample is representative of a single schoolboy team aged 15–18 years and may not represent rugby league programs tailored for other samples involving different playing levels, age groups, geographical regions, or sex in the school setting; therefore, we encourage similar investigation in other samples to widen the scope of evidence on this topic, which may also allow data to be reported specific to the playing position. Secondly, we quantified changes in performance using the most frequently adopted tests and testing protocols targeting specific physical qualities identified in the literature for this population [14]. Accordingly, this approach omits some physical qualities (e.g., flexibility [31], agility [32]) and tests (e.g., isometric midthigh pull [33], 1200 m time trial [2]) that may have gained more recent interest in the literature or demonstrated strong uptake in practice but not research. Moreover, variations in testing results that strictly focused on physical qualities may not translate to actual in-game performance. Thirdly, we drew inferences only between training loads and changes in testing outcomes indicative of physical qualities across rather acute periods (i.e., 4 weeks), and other factors (e.g., nutritional, psychological) may have contributed to the test performance at each time-point. Finally, the mesocycles we examined are not representative of the entire pre-season or in-season phases, with further investigation encouraged to elucidate more longitudinal trends in training load and changes in performance. 

## 5. Conclusions

Our study is the first to quantify the training loads of adolescent, schoolboy rugby league players, providing foundation data for this population. Of note, the observed weekly training loads are in line with other data reported for adolescent athletes in school-based sports programs [4], as well as adult, professional, rugby league players [22], indicating that these demands may be regularly experienced in school settings and potentially excessive for the age and level of athletes. Indeed, the training loads we observed corresponded with improved performance (i.e., physical qualities) across the 4-week pre-season mesocycle, but declines across the 4-week in-season mesocycle, suggesting they may be suitable for physical development in the short-term but detrimental when continued across the season. Regarding trends in training loads, we found that players experienced significantly higher sRPE loads, but similar session frequencies and durations accumulated across mesocycles during the pre-season compared to the in-season, with no clear week-to-week periodization strategy in each mesocycle within the school setting. 

## Figures and Tables

**Figure 1 sports-12-00251-f001:**
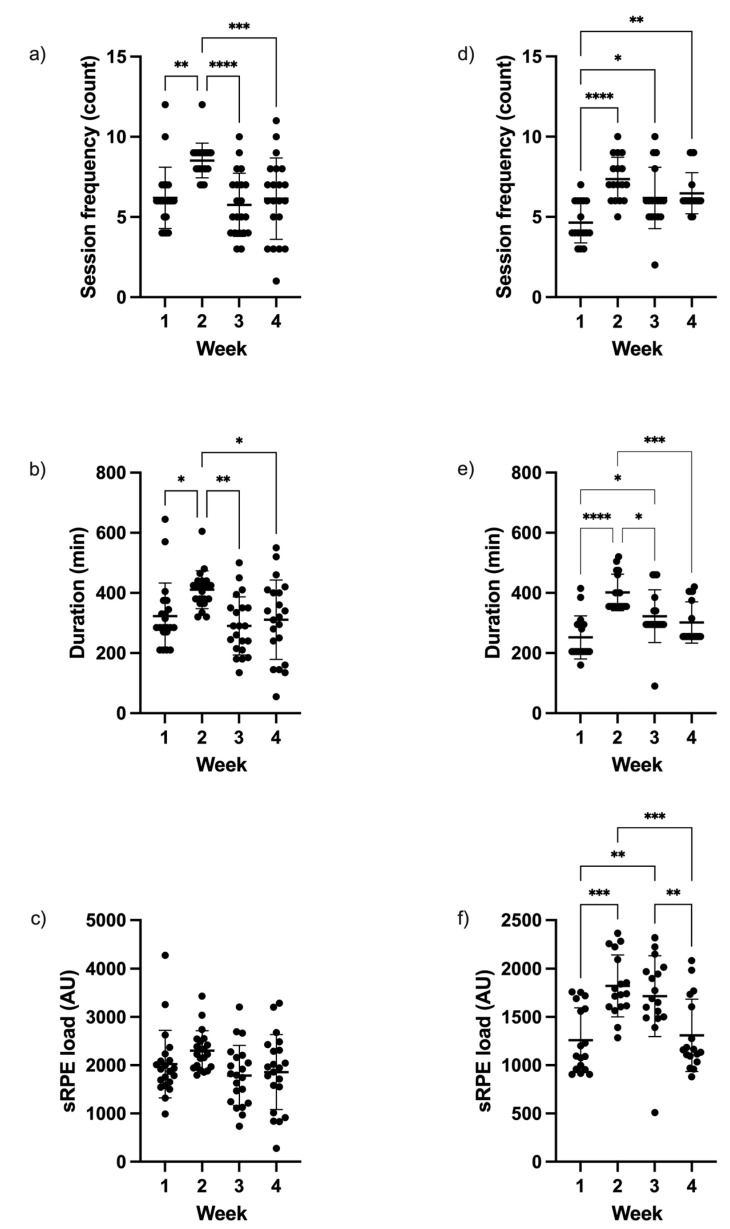
Mean ± standard deviation and individual data for (**a**) session frequency, (**b**) session duration, and (**c**) session rating of the perceived exertion (sRPE) load for each week during the pre-season, and (**d**) session frequency, (**e**) session duration, and (**f**) sRPE load for each week during the in-season in adolescent, schoolboy rugby league players. *Note*: Significant difference between weeks indicated by asterisks where * denotes *p* < 0.05, ** denotes *p* < 0.01, *** denotes *p* < 0.001, and **** denotes *p* < 0.0001.

**Table 1 sports-12-00251-t001:** Descriptive data and comparison statistics for session frequency, duration, and session rating of perceived exertion (sRPE) load between 4-week mesocycles monitored during the pre-season and in-season in adolescent, schoolboy, rugby league players.

Variable	n	Pre-Season	In-Season	Mean Difference (95% CI)	d_av_ (95% CI)	*p*
Sessions (count)	16	27.5 ± 6.3	24.9 ± 4.0	−2.6 (−5.9 to 0.8)	−0.32 (−0.62 to −0.01)	0.126
Duration (min)	16	1396 ± 330	1279 ± 82	−117 (−270 to 37)	−0.28 (−0.57 to 0.00)	0.127
sRPE load (AU)	16	8260 ± 2021	6148 ± 980	−2112 (−3030, to −1196)	−0.94 (−1.29 to −0.58)	<0.001 *

*Note*: Data presented as the mean ± standard deviation; * denotes significant difference between phases. *Abbreviations*: CI, confidence intervals; AU, arbitrary units.

**Table 2 sports-12-00251-t002:** Descriptive data and comparison statistics for physical qualities assessed at the baseline and following each 4-week mesocycle during the pre-season and in-season in adolescent, schoolboy, rugby league players.

Physical Quality	n	Time Point		Mean Difference (95% CI)	d_av_ (95% CI)	*p*
		Before	After			
Pre-season						
10 m linear sprint time (s)	16	1.854 ± 0.119	1.722 ± 0.058	0.132 (0.07 to −0.19)	−1.42 (−2.17 to −0.63)	<0.001 *
20 m linear sprint time (s)	16	3.130 ± 0.183	2.978 ± 0.212	0.152 (0.02 to −0.33)	−0.77 (−1.61 to 0.10)	0.083
505-Agility Test time (s)	16	2.349 ± 0.185	2.348 ± 0.116	0.001 (−0.11 to 0.11)	−0.01 (−0.66 to 0.64)	0.972
L-run Test (s)	16	5.808 ± 0.380	5.697 ± 0.186	0.111 (−0.136 to 0.359)	−0.37 (−1.14 to 0.40)	0.351
MSFT estimated VO_2max_ (mL·kg^−1^·min^−1^)	16	44.2 ± 3.7	47.5 ± 3.8	−3.3 (−5.3 to −1.4)	0.88 (0.30 to 1.44)	0.002 *
MBT distance (m)	16	6.86 ± 0.74	7.47 ± 0.61	−0.6 (−0.95 to −0.271)	0.90 (0.33 to 1.46)	0.002 *
CMJ height (cm)	16	63.8 ± 6.1	66.1 ± 6.0	−2.3 (−6.82 to 2.07)	0.40 (−0.30 to 1.07)	0.273
1-RM back squat (kg)	16	115.6 ± 11.4	133.5 ± 18.1	−17.9 (−24.00 to −11.75)	1.18 (0.61 to 1.74)	<0.001 *
1-RM bench press (kg)	16	90.9 ± 14.2	104.1 ± 13.3	−13.1 (−17.46 to −8.79)	0.96 (0.50 to 1.39)	<0.001 *
1-RM prone pow (kg)	16	80.9 ± 12.5	85.3 ± 12.5	−4.4 (−12.68 to 3.93)	0.35 (−0.28 to 0.97)	0.279
**In-season**						
10 m linear sprint time (s)	16	1.775 ± 0.136	1.803 ± 0.081	−0.028 (−0.07 to 0.17)	0.25 (−0.13 to 0.62)	0.202
20 m linear sprint time (s)	16	2.967 ± 0.132	2.987 ± 0.154	−0.020 (−0.05 to 0.01)	0.14 (−0.09 to 0.36)	0.230
505-Agility Test time (s)	16	2.124 ± 0.108	2.277 ± 0.303	−0.153 (−0.27 to −0.03)	0.67 (0.12 to 1.21)	0.016 *
L-run Test time (s)	16	5.559 ± 0.255	5.640 ± 0.173	−0.081 (−0.15 to −0.01)	0.37 (0.04 to 0.68)	0.027 *
MSFT estimated VO_2max_ (mL·kg^−1^·min^−1^)	16	49.1 ± 4.2	48.3 ± 3.5	0.8 (−0.7 to 2.4)	−0.22 (−0.59 to 0.16)	0.263
MBT distance (m)	15	6.84 ± 0.60	6.72 ± 0.49	0.12 (−0.30 to 0.26)	−0.22 (−0.47 to 0.05)	0.111
CMJ height (cm)	15	64.2 ± 4.2	62.4 ± 5.2	1.8 (−0.00 to 3.60)	−0.38 (−0.75 to 0.00)	0.050
1-RM back squat (kg)	16	138.1 ± 14.3	133.4 ± 14.7	4.7 (2.03 to 7.35)	−0.32 (−0.53 to −0.12)	0.002 *
1-RM bench press (kg)	16	105.3 ± 13.7	103.8 ± 13.5	1.5 (−0.96 to 4.08)	−0.11 (−0.29 to 0.06)	0.206
1-RM prone pow (kg)	16	88.4 ± 11.8	86.3 ± 10.1	2.1 (−0.38 to 4.76)	−0.19 (−0.42 to 0.03)	0.090

*Note*: Data presented as mean ± standard deviation; * denotes significant difference between phases. *Abbreviations*: CI, confidence intervals; MSFT, Multistage Fitness Test; MBT, medicine ball throw; CMJ, countermovement jump; and 1-RM, one-repetition maximum.

## Data Availability

The data that support the findings of this study are available from the corresponding author upon reasonable request, due to the protection of privacy of those who participated in this study.

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
