# Peer review of "Quantifying the Training Loads and Corresponding Changes in Physical Qualities among Adolescent, Schoolboy Rugby League Players"

_sports, 2024, doi:10.3390/sports12090251_

Round 1
Reviewer 1 Report
Comments and Suggestions for Authors
Dear authors.
Congrats for the research titled “Quantifying the Training Loads and Corresponding Changes in Physical Qualities among Adolescent, Schoolboy Rugby League Players”. It is an interesting research; however, it is necessary to take into account some recommendations so that the manuscript can be improved:
Abstract:
It is recommended to include a brief background to introduce the topic of study.
Introduction:
It is recommended to introduce a paragraph on why school rugby league players are an important cohort to study.
Materials and Methods:
It is necessary to enter information related to the informed consent signed by the legal guardians of the minors, as well as information on the approval of the research by an ethics committee.
It is recommended to introduce physical tests within a section such as Physical condition Test (for example)
Results:
The results obtained in the study are included, organizedly.
Discussion:
L 311-318: It is necessary to explain these lines. The authors indicate that there should be communication between coaches and teachers to try to balance workloads. However, it must be justified which is more important, or significant, if the sporting aspect or the educational aspect for this population.
Conclusions:
Adequate, oriented to the proposed objectives.
References:
Ok.
With the application of these changes, the quality of the manuscript will be improved.
Thank you
Author Response
Dear Reviewer,
Thank you for your comments on our manuscript and for providing an opportunity to revise our submission. We are grateful to you for the constructive comments. We have considered all the comments, and are confident that this manuscript has been improved as a result of the changes made.
Please find below our detailed responses to your comments and corresponding changes to the manuscript where applicable. Within the revised manuscript, new content is displayed as red text.
Comment 1:
Congrats for the research titled “Quantifying the Training Loads and Corresponding Changes in Physical Qualities among Adolescent, Schoolboy Rugby League Players”. It is an interesting research; however, it is necessary to take into account some recommendations so that the manuscript can be improved:
Response 1:
Thank you for your kind words. We are delighted that you took the time to review our manuscript and appreciate your efforts. We have considered all your comments and believe the resulting amendments have improved our manuscript greatly.
Comment 2:
Abstract: It is recommended to include a brief background to introduce the topic of study.
Response 2:
Thank you for your comment, we agree that a brief background to introduce the topic would improve the abstract. We were previously constrained by the word count; however, we have now addressed this aspect by including a brief introduction.
Action 2:
Please see the amendment to lines 17-22: “The adolescence development period is critical for rugby league athletes given the physical growth, neuromuscular adaptation, and skill acquisition that occur. Secondary schools play an important role in the development of adolescent rugby league players; however, players may be selected into rugby league academies and development programs outside of school, as well as participate in additional sports. In turn, the training loads these young athletes accrue and implications of these loads are currently unknown.”
Comment 3:
Introduction: It is recommended to introduce a paragraph on why school rugby league players are an important cohort to study.
Response 3:
Thank you for your comment. We agree that the introduction of a paragraph on why school rugby league players are an important cohort to study will improve the manuscript.
Action 3:
Please see the amendment to lines 53-59: “Schoolboy, adolescent rugby league players are at a particularly important time in their athletic development. Given the potential for adolescents to progress to professional ranks, Australian organisations like the Queensland Rugby League [2] support school systems that can have a considerable influence on the development of players. Indeed, with ~240,000 students engaging with rugby league in the Australian school setting [1], school systems have the capacity to influence a considerable number of developing players.”
References (corresponding with in text citation number):
- National Rugby League. National Rugby League annual report 2022. https://www.nrl.com/siteassets/about/annual-reports/nrl-gen22_7113-annual-report-2022_lr.pdf. 2022, April 7.
- Queensland Rugby League. Queensland Rugby League Physical Performance Framework. https://www.qrl.com.au/contentassets/49095db3d85a43c0b164988fbb4f158f/qrl21_physicalperformanceframework_digital_final2.pdf. 2020, April 7.
Comment 4:
Materials and Methods: It is necessary to enter information related to the informed consent signed by the legal guardians of the minors, as well as information on the approval of the research by an ethics committee.
Response 4:
We thank you for your comment. We absolutely agree, and uphold sound ethical practice. We refer to our statement presented in lines 128-131: “Written informed assent and consent were obtained from all players and their legal guardians prior to participation. Ethical approval for this study was obtained from the institutional Human Research Ethics Committee (#0000023570).”
Action 4:
A statement was present which addressed the comment received, therefore, no action was required.
Comment 5:
It is recommended to introduce physical tests within a section such as Physical condition Test (for example)
Response 5:
Thank you for this recommendation. Introducing the implemented tests within the methods section is certainly valuable for the reader. Our current subsection heading 2.4 titled “Physical Testing” may not be sufficient as you suggest. “Physical condition” is not a term consistent with the literature cited within our manuscript; however, we have revised the title of this section based on your comment.
Action 5:
Please see the amendment to line 150: “Tests Used to Assess Physical Qualities”
Comment 6
Results: The results obtained in the study are included, organizedly.
Response 6:
We thank you for recognizing the organization and presentation of our findings.
Comment 7.
Discussion: L 311-318: It is necessary to explain these lines. The authors indicate that there should be communication between coaches and teachers to try to balance workloads. However, it must be justified which is more important, or significant, if the sporting aspect or the educational aspect for this population.
Response 7:
Thank you for this comment. We certainly understand why you suggest that there must be justification for which is more important – the sporting or educational aspect. However, this notion is not the intent of this statement and therefore we have added further detail to better articulate our message. Our statement explicitly refers to the sporting aspect (and not educational requirements or performance), for which we are eluding to communication between the school team coaching staff and the external team coaching staff.
Action 7:
Please see the amendment to lines 330-331: “…communication between school team coaching staff and external team coaching staff (e.g., academy, representative teams)…”
Comment 8:
Conclusions: Adequate, oriented to the proposed objectives.
Response 8:
We thank you for the time spent reviewing our conclusion and the positive appraisal given.
Comment 9:
References: Ok.
Response 9:
We thank you for the time spent reviewing our references.

Reviewer 2 Report
Comments and Suggestions for Authors
The proposed article reveals important issues of evaluation the physical activity in the process of training of young athletes. The relevance of this article was increased due to the increased quantity of activity in the process of training of young athletes in rugby, revealed in the course of the research.
The article was structured logically, all stages of the research were described in detail. The subject for additional analysis may be the periods of research chosen by the author, why exactly these stages and the total duration of the research were observed.
Regarding specific comments on the matter of the article:
• it should be clarified the use of abbreviations (AU) in the text of the article;
• VO2max [mL·kg-1 ·min-1 ] was evaluated during the Aerobic Capacity analysis, it is necessary to clarify whether it is possible to apply this technique for the selected contingent of the subjects of research. In the proposed link (https://www.ncbi.nlm.nih.gov/pmc/articles/PMC1478728/?page=4), the data of the study were implemented for the contingent of persons aged 19-36;
• the title of subsection 2.4. should be revises, because of not corresponding to the content;
• the principle of choosing the tests for evaluation the physical condition of athletes should be specified, why was the evaluation of flexibility not carried out?
• an interesting direction of research could have been conducting an analysis in accordance with the playing role of athletes; the authors should have taken into account this direction of research in accordance with the prospects of future research or limiting factors;
• it is better to present the data for each of the periods separately without averages output in the lines 273 - 274;
• it is necessary to clarify which in-season mesocycle was taken for analysis, it is necessary to characterize the intensity of competitive activity in this period, which can significantly affect the obtained conclusions on the study.
Author Response
Dear Reviewer,
Thank you for your comments on our manuscript and for providing an opportunity to revise our submission. We are grateful to you for the constructive comments. We have considered all the comments, and are confident that this manuscript has been improved as a result of the changes made.
Please find below our detailed responses to your comments and corresponding changes to the manuscript where applicable. Within the revised manuscript, new content is displayed as red text.
Reviewer 2
Comment 1:
The proposed article reveals important issues of evaluating the physical activity in the process of training of young athletes. The relevance of this article was increased due to the increased quantity of activity in the process of training of young athletes in rugby, revealed in the course of the research.
The article was structured logically, all stages of the research were described in detail. The subject for additional analysis may be the periods of research chosen by the author, why exactly these stages and the total duration of the research were observed.
Response 1:
We thank you for the time spent reviewing our manuscript and are grateful for your comments.
Comment 2:
It should be clarified the use of abbreviations (AU) in the text of the article.
Response 2:
Thank you for identifying this oversight. We have now defined the abbreviation AU as arbitrary units within both the abstract and main body of the manuscript.
Action 2:
Please see the amendment to lines 31, and 83-84: “… arbitrary units [AU]) …”
Comment 3:
VO2max [mL·kg-1 ·min-1] was evaluated during the Aerobic Capacity analysis, it is necessary to clarify whether it is possible to apply this technique for the selected contingent of the subjects of research. In the proposed link (https://www.ncbi.nlm.nih.gov/pmc/articles/PMC1478728/?page=4), the data of the study
were implemented for the contingent of persons aged 19-36;
Response 3:
Thank you for your comment. The main approach within our methodology involves utilising the tests and testing protocols most frequently adopted within studies examining adolescent, male rugby league players. This approach is based on a recently published, comprehensive review of the literature (Carron et al., 2023). In this regard, we highlight this approach on lines 151–152 and indicate the review identified the Multistage Fitness Test using procedures published by Ramsbottom et al. (1988) as the most common test and protocol to assess and predict aerobic capacity in this population. We recognise that Ramsbottom et al. (1988) investigated participants aged 19–36 years, and that an individual’s age is not factored into the formula to estimate VO2max. However, this approach is the most commonly used equation to estimate VO2max in adolescent, male rugby league players, who are on the verge of the age range initially validated, and producing estimated VO2max values in the range of those expected for an adult male population (Gabbett, 2000; Gabbett 2002).
References:
Carron MA, Scanlan AT, Power CJ, Doering TM. What tests are used to assess the physical qualities of male, adolescent rugby league players? a systematic review of testing protocols and reported data across adolescent age groups. Sports Med Open. 2023;9(106). doi: 10.1186/s40798-023-00650-z
Ramsbottom R, Brewer J, Williams C. A progressive shuttle run test to estimate maximal
oxygen uptake. Br J Sports Med. 1988;22(4):141-4.
Gebbett T. Physiological and anthropometric characteristics of amateur rugby league players.
Br J Sports Med. 2000;34:303-307.
Gabbett T. Physiological characteristics of junior and senior rugby league players. Br J Sports
Med. 2002;36(5):334-339. doi: 10.1136/bjsm.36.5.334
Action 3:
No action taken.
Comment 4:
The title of subsection 2.4. should be revised, because of not corresponding to the content;
the principle of choosing the tests for evaluation of the physical condition of athletes should be specified, why was the evaluation of flexibility not carried out?
Response 4:
We thank you for your comment concerning the title of subsection 2.4. Upon reflection, we agree that the title should more appropriately correspond to the content, and have changed it based on your suggestion and the comments provided by Reviewer 1.
Concerning testing for flexibility, we based our testing battery on the most commonly adopted tests and testing protocols used to assess physical qualities among adolescent rugby league players, as identified in a recent systematic review (Carron et al., 2023) and specified on lines 151–152. Following this methodology, flexibility tests were not identified as being regularly adopted in this previous review and therefore not included in our testing battery. However, we agree that this approach may bring some limitations to the work, including the omission of certain physical qualities like flexibility, which should be acknowledged.
References (corresponding with in text citation number):
Carron MA, Scanlan AT, Power CJ, Doering TM. What tests are used to assess the physical
qualities of male, adolescent rugby league players? a systematic review of testing protocols
and reported data across adolescent age groups. Sports Med Open. 2023;9(106). doi:
10.1186/s40798-023-00650-z
Action 4:
Please see the amended title on line 150: “Tests Used to Assess Physical Qualities”.
Moreover, the following statement has been added to lines 379–382 to acknowledge the scope of testing included: “Accordingly, this approach omits some physical qualities (e.g., flexibility [31], agility [32]) and tests (e.g., isometric midthigh pull [33], 1,200-m time trial [2]) that may have gained more recent interest in the literature or demonstrated strong uptake in practice but not research.”
Comment 5:
An interesting direction of research could have been conducting an analysis in accordance with the playing role of athletes; the authors should have taken into account this direction of research in accordance with the prospects of future research or limiting factors.
Response 5:
Thank you for this comment. We agree that positional differences are a key consideration and certainly an area of interest. However, given the relatively small sample size recruited in our study from a single school, splitting the sample for such analyses was not possible. Nevertheless, future work should examine the specific loads and changes in physical qualities across timepoints throughout the season according to playing position and we have updated our limitations section to reflect this thought.
Action 5:
Please see the amendment to lines 373-375: “… we encourage similar investigation in other samples to widen the scope of evidence on this topic, which may permit analyses specific to playing positions.”
Comment 6:
It is better to present the data for each of the periods separately without averages output in the lines 273 – 274.
Response 6:
Thank you for raising this thought-provoking question. However, we believe that it is most appropriate to report the average weekly training loads encountered separately across each mesocycle to summarise this part of our study. In turn, reporting wider data here may be too extensive and repetitive (from the results) for an effective discussion section. Our current approach is important given it remains consistent with the format in which training load data are typically reported in the literature within similar cohorts, which allows for direct comparison of findings (Scantlebury et al (2021); Hartwig et al., (2008); and Coutts et al., (2007)). Furthermore, our load data are presented in a weekly fashion within Figure 2, and our statistical comparisons were performed between weeks, which creates strong fluency across sections in summarizing our data this way. We have, however, reported the average load data across each entire mesocycle in Table 1 should this information be sought.
References:
Scantlebury ST, K., Saqszuk, T., Dolton-Barron, N., Phibbs, P., Jones, B. The frequency and
intensity of representative and nonrepresentative late adolecent team-sport athletes’ training
schedules. J Strength Cond Res. 2021;1(12):3400-6. doi: 10.1519/JSC.0000000000003449.
PMID: 32084108.
Hartwig T, Naughton G, Searl J. Defining the volume and intensity of sport participation in
adolescent rugby union players. Int J Sports Physiol Perform. 2008;3(1):94-106. doi:
10.1123/ijspp.3.1.94.
Coutts AJ, Reaburn P, Piva TJ, Roswell GJ. Monitoring for overreaching in rugby league
players. Eur J Appl Physiol. 2007;99(3):313-24. doi: 10.1007/s00421-006-0345-z.
Action 6:
No action taken.
Comment 7:
It is necessary to clarify which in-season mesocycle was taken for analysis, it is necessary to characterize the intensity of competitive activity in this period, which can significantly affect the obtained conclusions on the study
Response 7:
We agree with your line of thought and appreciate your attention to detail in pointing out the need for further information regarding the mesocycles monitored. In this regard, we monitored the final mesocycle during the in-season phase, which warrants mentioning. Based on your comment, we thought it is necessary to provide the same details for the pre-season cycle.
Action 7:
Please see the amendment to lines 112-115: “The monitored pre-season mesocycle comprised the last 4 weeks (out of a 6-week pre-season phase), while the monitored in-season mesocycle comprised the last 4 weeks (out of a 12-week in-season phase) and contained the final four games of the competitive schoolboy rugby league season.”

Round 2
Reviewer 1 Report
Comments and Suggestions for Authors
Thank you for taking my comments into account